# Quantitative Risk Assessment for the Introduction of Bovine Leukemia Virus-Infected Cattle Using a Cattle Movement Network Analysis

**DOI:** 10.3390/pathogens9110903

**Published:** 2020-10-28

**Authors:** Kosuke Notsu, Anuwat Wiratsudakul, Shuya Mitoma, Hala El Daous, Chiho Kaneko, Heba M. El-Khaiat, Junzo Norimine, Satoshi Sekiguchi

**Affiliations:** 1Graduate School of Medicine and Veterinary Medicine, University of Miyazaki, Miyazaki 889-1692, Japan; notsu.kousuke.p0@cc.miyazaki-u.ac.jp (K.N.); gf11027@student.miyazaki-u.ac.jp (S.M.); hala.ali@fvtm.bu.edu.eg (H.E.D.); 2Department of Clinical Sciences and Public Health, Faculty of Veterinary Science, Mahidol University, Nakhon Pathom 73170, Thailand; anuwat.wir@mahidol.edu; 3The Monitoring and Surveillance Center for Zoonotic Diseases in Wildlife and Exotic Animals, Faculty of Veterinary Science, Mahidol University, Nakhon Pathom 73170, Thailand; 4Faculty of Veterinary Medicine, Benha University, Toukh 13736, Egypt; elkaiatetal2009@yahoo.com; 5Center for Animal Disease Control, University of Miyazaki, Miyazaki 889-2192, Japan; ckaneko@cc.miyazaki-u.ac.jp (C.K.); nori@cc.miyazaki-u.ac.jp (J.N.); 6Department of Veterinary Science, Faculty of Agriculture, University of Miyazaki, Miyazaki 889-2192, Japan

**Keywords:** bovine leukemia virus, enzootic bovine leukosis, animal movement network analysis, cattle introduction, quantitative risk assessment

## Abstract

The cattle industry is suffering economic losses caused by bovine leukemia virus (BLV) and enzootic bovine leukosis (EBL), the clinical condition associated with BLV infection. This pathogen spreads easily without detection by farmers and veterinarians due to the lack of obvious clinical signs. Cattle movement strongly contributes to the inter-farm transmission of BLV. This study quantified the farm-level risk of BLV introduction using a cattle movement analysis. A generalized linear mixed model predicting the proportion of BLV-infected cattle was constructed based on weighted in-degree centrality. Our results suggest a positive association between weighted in-degree centrality and the estimated number of introduced BLV-infected cattle. Remarkably, the introduction of approximately six cattle allowed at least one BLV-infected animal to be added to the farm in the worst-case scenario. These data suggest a high risk of BLV infection on farms with a high number of cattle being introduced. Our findings indicate the need to strengthen BLV control strategies, especially along the chain of cattle movement.

## 1. Introduction

Bovine leukemia virus (BLV) belongs to the genus *Deltaretrovirus* of the *Retroviridae* family, and it is the causative agent of enzootic bovine leukosis (EBL). Although most infected cattle are asymptomatic carriers, 1% to 5% develop fatal lymphosarcoma several years after infection [1,2]. In addition, BLV infection leads to high economic losses in the dairy and beef industries due to reduced milk production in infected cattle, lower fertility rates, and cattle culling or death [3,4,5].

BLV infection has a worldwide distribution, and EBL was listed by the World Organization for Animal Health (OIE) as a disease that could significantly impact international trade [6]. According to previous studies, animal-level seroprevalence of BLV infection is 38.6% in the United States [7], 18.29% in China [8], 2.28% in Turkey [9], and 0.04% in Italy [10]. In Japan, EBL is a notifiable disease and has been subjected to passive surveillance since 1997 [11]. From 2009 to 2011, the animal-level prevalence of BLV infection in the country was 35.2% [12].

BLV transmission in animal populations occurs via the transfer of infected lymphocytes from BLV-infected animals [13,14]. The potential of within-farm transmission is mainly associated with farming practices that involve the contamination of infected blood, such as the repeated use of contaminated needles, dehorning, and rectal palpation with a common sleeve. In addition, direct contact between BLV-infected and BLV-free animals was also identified as a risk factor for BLV transmission [15]. In Japan, the presence of horseflies in the summer was associated with within-herd transmission of BLV [16]. Nevertheless, for between-herd transmission, cattle introduction from other farms is the main risk factor [8,9,17,18]. However, although the risk of adding BLV-infected cattle onto a farm is positively associated with the total number of cattle introduced into a herd, the quantity of this risk is unclear.

To estimate the risk of disease transmission through between-herd movement of livestock, researchers in the field of veterinary epidemiology have applied a social network analysis (SNA) [19]. SNA is widely used for exploring disease diffusion by the movement of animals. This method leads to accurate risk assessments and better control measures, including active surveillance and intervention [20,21,22,23,24,25,26,27,28]. For example, SNA was used to study the cattle trade in a northern province of Thailand [25], where the network was used to examine different market closure interventions to control foot-and-mouth disease [23]. In addition, SNA has been exploited to investigate the movements of other animal species such as horses [26], pigs [27], and chickens [28]. SNA is also applicable to the analysis of disease introduction at the farm level, as SNA focuses on the links between different sources and directions. Therefore, the objective of this study was to employ SNA to quantitatively estimate the risk of the introduction of BLV-infected cattle through the chains of cattle movement.

## 2. Results

### 2.1. Proportion of BLV-Positive Cattle

In total, 997 blood samples of cattle were collected; 457 samples were collected from slaughterhouse A and the remaining were collected from slaughterhouse B. Of note, we excluded 153 samples from our study as the movement history for these animals was not available (positive: 40, negative: 113). Out of the 844 remaining samples, 194 (23.5%) were positive on a BLV-enzyme-linked immunosorbent assay (BLV-ELISA) test (Table 1). The rate of positives was 25.8% in males and 21.8% in females. The proportion of positive animals among Japanese black, Holstein, and F1 cattle was 22.3%, 66.7%, and 34.6%, respectively.

### 2.2. Cattle Movement Network Analysis

Based on the movement history of 844 traced cattle, 1097 farms, 55 markets, and 2 slaughterhouses, a total of 2963 movements (farm–farm, farm–market, market–farm, farm–slaughterhouse, and market–slaughterhouse) were identified. A non-weighted cattle movement network was constructed (Appendix A). As a result of the conversion from a non-weighted to a weighted network, a total of 1641 directed links between nodes were observed. We modeled two scenarios, the worst-case scenario (scenario 1) and the most likely scenario (scenario 2). In scenario 1, 145 farms (13.22%) out of 1097 farms were classified as being infected cattle introduced (Int) farms, and the rest were classified as infected cattle non-introduced (N-Int) farms. In scenario 2, 66 farms (6.02%) were classified as Int farms. Based on the farm statuses (Int farm or N-Int farm) and the weight of the ties on the directed links between nodes, a scenario-based network of cattle movement was constructed (Figure 1).

In both scenarios, 663 N-Int farms out of 1097 total farms were classified as birth farms and thus were excluded from our statistical analysis in order to focus the analysis on cattle introduction along the movement routes. Comparative tests using 434 farms (scenario 1: 145 Int farms and 289 N-Int farms and scenario 2: 66 Int farms and 368 N-Int farms) suggested that Int farms had a higher value of weighted in-degree centrality than N-Int farms (scenario 1: *p* < 1 × 10^−12^ and scenario 2: *p* < 1 × 10^−10^), as shown in Figure 2. Therefore, the number of introduced cattle was a significant risk factor for being an Int farm in both scenarios.

### 2.3. Estimation of the Number of BLV-Infected Cattle Among Introduced Cattle

On the basis of the comparative test of weighted in-degree centrality, 663 birth farms were excluded from the 1097 total farms. Among the remaining 434 farms (scenario 1: 145 Int farms and 289 N-Int farms and scenario 2: 66 Int farms and 368 N-Int farms), a histogram indicating the number of farms relative to the weighted in-degree centrality was constructed to show the distribution of sample number (Appendix A). The number of samples from the farms tended to concentrate at the lower value of weighted in-degree centrality. Thus, 21 Int farms with weighted in-degree centrality over nine were excluded to avoid bias from having too few samples. In total, 413 farms (scenario 1:124 Int farms and 289 N-Int farms and scenario 2:53 Int farms and 360 N-Int farms) were included in this analysis. A generalized linear mixed model was constructed to describe the association between weighted in-degree centrality and the number of introduced infected cattle. The fit plot (Figure 3) of weighted in-degree centrality and the estimated number of introduced BLV-infected cattle indicated a positive association between these two variables in both scenarios (scenario 1: *p* < 1 × 10^−12^ and scenario 2: *p* < 1 × 10^−2^).

## 3. Discussion

The introduction of cattle from other farms was previously identified as a significant risk factor for within-farm transmission of BLV [8,9,17,18]. We used an online official record to trace the movement of individual cattle. We then quantitatively analyzed the risk of BLV-infected cattle introduction at the farm level through the chains of cattle movements. We employed a cattle movement network analysis that took into account the infectious status of all involved cattle. Previously, network analyses have been used to assess the dissemination of infectious diseases, such as brucellosis, foot-and-mouth disease, and classical swine fever, and to propose effective interventions [20,23,27,29,30]. To our knowledge, this is the first report of a network analysis applied to the quantitative risk assessment of BLV infection.

As expected, weighted in-degree centrality was positively associated with the number of introduced BLV-infected cattle in both scenarios. However, the introduction risk of BLV-infected cattle differed on the basis of the scenario. On the farm that introduced six cattle from outside annually, one BLV-infected animal was introduced per year in scenario 1 (Figure 3a). In scenario 2, one infected animal was introduced every five years (0.2 infected cattle per year, Figure 3b). These data indicate the importance of BLV control on birth farms.

The cattle that tested positive on the BLV-ELISA were assumed to be infected with BLV from their birth farms, and their infectious status was sustained until slaughter. This scenario was used to estimate the highest number of BLV-infected cattle among introduced cattle on each farm. In order to better understand disease transmission and establish interventions, the fundamental first approach is to assume the worst-case scenario [31,32,33]. Even if we could not clearly determine when and where the cattle were infected, the cattle movement network in which cattle infectious status was identified at slaughterhouses allowed us to predict the farm-level risk of BLV introduction under this scenario. As it is not obligatory to implement any interventions at the farm level to prevent and control BLV, taking samples at slaughterhouses is more appropriate in terms of biosecurity. Thus, our approach provides a simple and practical platform for the risk estimation of BLV introduction. This method is promptly available for use in different settings where persistent infectious diseases are identified.

In this study, we found an important risk factor for BLV introduction at the farm level. A higher chance of introducing BLV-infected cattle was observed on the farms with higher weighted in-degree centrality. Thus, the prioritization of these farms as the targets for BLV surveillance is suggested to better control disease spread. Similarly, Stärk et al. (2006) demonstrated that risk-based surveillance provided a higher probability for disease detection with a higher cost-efficacy [34]. We are interested in increasing the efficacy of detection and removal of BLV-infected cattle using diagnostic tests and the restriction of cattle export from farms with higher weighted in-degree centrality for BLV elimination. The assessment of these control strategies with a dynamic mathematical modeling of BLV spread on cattle movement networks is also recommended. Previous studies using mathematical simulations for other diseases such as brucellosis on cattle movement networks found that canceling outgoing movements of cattle from nodes with the highest value of degree centrality (top 1%) or in-degree centrality (top 2%) resulted in the highest reduction of infected nodes [29,35]. Such applications are helpful to improve the risk assessment at the farm level and develop relevant risk-based surveillance and control strategies.

Movements of infected animals must be considered for the control of chronic infectious diseases in livestock. Nekouei et al. (2016) reported that BLV-infected cows with two and three lactations showed significantly decreased lifetime milk production (3609 kg to 1500 kg and 2051 kg to 292 kg, respectively) compared with their BLV-negative counterparts, even if the infected animals never developed EBL [4]. In addition to BLV infection, Johne’s disease and bovine viral diarrhea (BVD) cause severe economic impacts in the cattle industry [3,4,5,36]. Cows in BVD-seropositive herds had reductions in 305-d milk, fat, and protein of 368 kg, 10.2 kg, and 9.5 kg, respectively, compared with cows in BVD-seronegative herds [36]. Regarding the pathogen implicated in Johne’s disease, cows in *Mycobacterium avium* subsp. *paratuberculosis* (MAP)-seropositive herds with more than 4 lactations had reductions in 305-d milk of 212 kg compared with their MAP-seronegative counterparts [36]. Thus, early removal of pathogens is financially beneficial to farmers. However, these diseases spread without the awareness of farmers and veterinarians due to unobvious clinical signs in infected animals. Once they have spread, a complete removal of pathogens from the field is not economically feasible. First, the transmission route of the pathogen via animal movements must be clarified, as the between-farm transmission of the disease mainly results from the introduction of infected animals [37]. Marquetoux et al. (2016) and Booth et al. (2013) demonstrated that a combination of cattle movement network analyses and phylogenic information from the isolated pathogens resulted in a better understanding of transmission routes for Johne’s disease and BVD [38,39]. Likewise, a network analysis based on field-sampled data is useful to identify the between-farm transmission of pathogens via the movement of infected animals and to further control the spread of chronic infectious diseases at a regional level.

BLV-infected cattle generate continuous anti-BLV antibodies throughout the course of their lives. This characteristic supports the usage of slaughterhouses as the basepoint of sampling, diagnosis, and epidemiological investigations. Indeed, BLV surveys targeting slaughtered animals were previously conducted to estimate the regional prevalence of BLV in the United States [7].

We acknowledge some limitations of this study. Epidemiological investigation based on sampling in slaughterhouses cannot determine when and where each animal was infected with BLV. Additionally, within-farm transmission of BLV was not considered.

In conclusion, using a cattle movement network analysis that considered BLV-infectious status allowed us to estimate the risk of BLV-infected cattle introduction on a between-farm basis. Our findings quantitatively suggested that farms with more cattle introduction were likely to introduce BLV into their herds. Our results indicate that a BLV control strategy focused on between-farm movement of cattle is crucially needed. Additionally, this study highlighted the importance of BLV control on birth farms. These risk-based approaches for assessment and control are very useful and efficient in not only BLV, but other animal infectious diseases.

## 4. Materials and Methods

### 4.1. Sample Collection and BLV Diagnostic Testing

#### 4.1.1. Blood Sample Collection

Blood sample collection was conducted in two slaughterhouses (slaughterhouse A and B) in Miyazaki prefecture, which is located on Kyushu Island in southern Japan between 32°03′ and 32°44′ N latitude and 131°42′ and 131°53′ E longitude (Figure 4). Blood samples were collected from December 2015 to June 2016 and from August to September 2016 in slaughterhouses A and B, respectively. We collected blood from the necks of the cattle during slaughter, and it was put in tubes containing EDTA (Ethylene-Diamine-Tetra-Acetic acid) by veterinarians. The samples were then centrifuged at 1500× *g* for 5 min for plasma separation and stored at −20 °C in the laboratory of the University of Miyazaki for serological tests. Data from all cattle, including sex, age, breed, carcass weight, and ear tag, were provided by the meat inspection office of each slaughterhouse and recorded in our database together with the date of sampling.

#### 4.1.2. BLV Serological Testing

Serum samples were examined with a commercial BLV gp51 antibody detection enzyme-linked immunosorbent assay (ELISA) kit (JNC Co., Ltd., Tokyo, Japan) for the presence of BLV antibodies. The test was performed according to the manufacturer’s instructions.

### 4.2. Cattle Movement Network Analysis

#### 4.2.1. Data Source

Data on cattle movements were retrieved from the Search Service of Individual Identification Information of Cattle managed by the National Livestock Breeding Center (NLBC) using ear tag numbers. The movement history of each animal included the identification and region for all farms, markets, packers, and slaughterhouses resided upon over the animal’s lifetime. The prefecture where the cattle were born, the slaughterhouse, slaughtering date, ELISA S/P (Sample to Positive) ratio, sex, age, breed, and carcass weight of all cattle are shown in the Appendix A.

#### 4.2.2. Construction of Cattle Movement Network

A static non-weighted directed network indicating lifelong movements of cattle was constructed with the package “igraph” [40] equipped in the program R (v. 3.6.2; R core team, Vienna, Austria). A node referred to each premise, including farms, markets, and slaughterhouses, and a directed tie indicated a direction of cattle movement. All nodes were distributed on the surface of the sphere uniformly [41]. 

Directed ties on the same source and direction along each node were synthesized to one tie and weighted by the number of directed ties on the same source and direction (Figure 5).

#### 4.2.3. Farm-Level Measures for Centrality Analysis

A degree is the number of nodes connected to the focal node [42]. Degree centrality is a measurement of the degree in any focal nodes in the network. A node with high-degree centrality is more connected to other nodes compared to a node with a low value. In the case of directed networks, we distinguished between in-degree (number of incoming ties) and out-degree (number of outgoing ties) of the nodes. In-degree centrality of the node *i* (*C_D_-_in_* (*i*)) was calculated with the following equation [42]:$$C_{D-in}\left(i\right)\;=\;k_{i}^{in}$$
where *k _i_^in^* denoted the number of nodes moving toward node *i*.

We extended our calculation of degree to the sum of the weight when analyzing weighted networks [42,43,44,45]. The weighted in-degree centrality (*C_D_-_in_^w^* (*i*)) was formalized as follows [42]:$$C_{D-in}^{w}\left(i\right)\;=\;k_{i}^{in}\;\times \;S_{i}^{in}$$
where *S_i_^in^* denoted the weight of the tie moving toward node *i*. The weight was calculated from the total number of moved cattle on each tie (Figure 5). To focus on the probability of contaminating BLV-infected cattle among introduced cattle, weighted in-degree centrality and the number of introduced BLV-infected cattle of each farm (presumed on the following scenario) were generated from the cattle movement network.

#### 4.2.4. Construction of a Scenario-Based Network of Cattle Movement

As we investigated the staying periods of all cattle on each farm, we found that a majority of their residence time was on their birth farm or first-moved farm (Appendix A). We constructed two scenarios based on the assumption that BLV-ELISA-positive cattle were infected at either of these farms. In scenario 1, BLV-ELISA-positive cattle were assumed to have been infected at their birth farms and their BLV infectious status remained positive until slaughtered. This was a worst-case scenario in which the highest number of BLV-infected cattle introduction onto each farm had occurred. Scenario 2 was constructed as the most likely scenario. We assumed that all sampled cattle were born from a different mother. Of those 198 BLV-ELISA-positive cattle, 42.4% of their mothers were assumed to be infected with BLV according to a previous nationwide survey [12]. Notably, the BLV prevalence of beef breeding cattle in the Kyushu/Okinawa region was used because a majority (93.9%) of BLV-ELISA-positive cattle were born in this region. According to a previous study, 18.6% of calves were assumed to have been transmitted BLV vertically from their BLV-infected mothers [46]. Thus, 7.89% (the 42.4% BLV prevalence of mothers multiplied by the vertical transmission rate of 18.6%) of randomly selected BLV-ELISA-positive cattle were assumed to have been infected on their birth farms, and the rest were assumed to have been infected on their first-moved farms. Under this scenario, farms involved in the cattle movement network were classified into two types, namely infected cattle introduced (Int) farms and infected cattle non-introduced (N-Int) farms. We followed the criteria proposed in a previous risk analysis using a cattle movement network and the infectious status of *Taenia saginata* diagnosed at slaughterhouses [47], in which an Int farm was defined as a farm that BLV-infected cattle were introduced into at least once, and such cattle have never been introduced into N-Int farms (Figure 5). A scenario-based network of cattle movement was constructed to visualize the relationship between weighted in-degree centrality and the status of the farms (Int or N-Int) (Figure 1).

### 4.3. Statistical Analysis

Wilcoxon’s rank test was used to assess the difference of weighted in-degree centrality between Int farms and N-Int farms to determine the risk of weighted in-degree centrality for being Int farms. *p* values < 0.05 were considered statistically significant.

A generalized linear mixed Poisson model was built to judge the effect of weighted in-degree centrality (dependent variable) on the number of introduced BLV-infected cattle (independent variable) using the “glmer” function in the “lme4” package [48]. The Poisson model was selected, as it was appropriate for modeling counts of relatively rare events (the number of introduced BLV-infected cattle, *γ_i_*). Farm-specific random effects (*R_ifarm_*) were incorporated into the model with a fixed effect for the weighted in-degree centrality (*X_i_*). The model is presented in the following form:$$Y_{i}\;\sim Poisson\;\left({ɤ}_{i}\right)$$
$$\gamma_{i}\;=\;\beta_{0}\;+\;\beta_{1}X_{i}\;+\;R_{ifarm}$$
where *i* denoted the farm and *β* was the fixed effect. A fit plot based on the generalized linear model was constructed using the “plot_model” function in the “sjPlot” package [49].

All statistical analyses were conducted in R (v. 3.6.2; R core team, Vienna, Austria).

### 4.4. Ethics Statement

The protocol used in this study was reviewed by the Cattle Ethics Committee of the University of Miyazaki’s Faculty of Agriculture.

## Figures and Tables

**Figure 1 pathogens-09-00903-f001:**
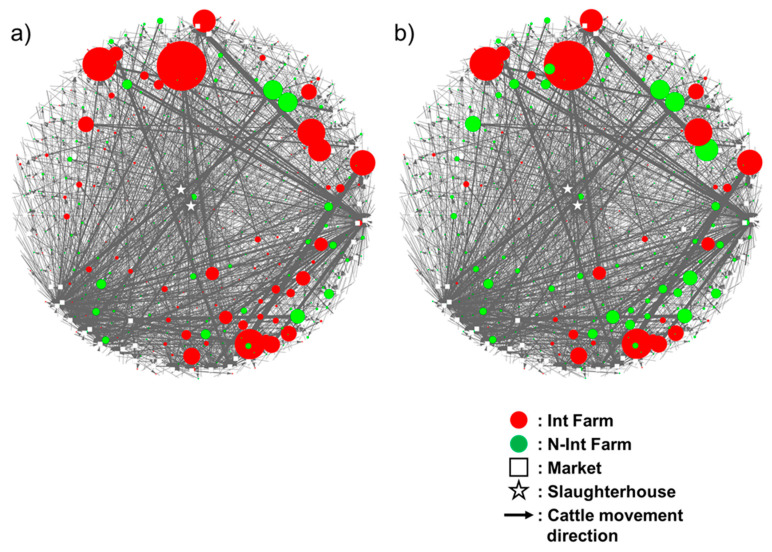
A scenario-based network of cattle movement. Subfigure (**a**) depicts the network based on scenario 1 and subfigure (**b**) depicts the network based on scenario 2. The networks involve 1097 farms, 55 markets, and 2 slaughterhouses, with 1,641 movements between premises (farm–farm, farm–market, market–farm, farm–slaughterhouse, and market–slaughterhouse). Nodes are depicted as farms (circles; red: infected cattle introduced (Int), green: infected cattle non-introduced (N-Int)), markets (white squares), and slaughterhouses (stars). A directed edge represents the cattle movement between premises, and the arrowhead indicates the direction of the movements. Each edge was weighted by the number of moved cattle, except for ties toward slaughterhouses. The size of each farm node reflects the value of weighted in-degree centrality.

**Figure 2 pathogens-09-00903-f002:**
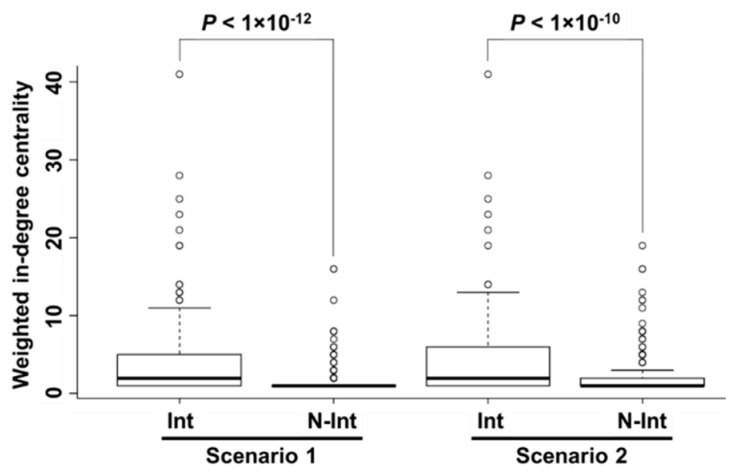
Comparison of weighted in-degree centrality between Int farms and N-Int farms in each scenario. A box and whisker plot of weighted in-degree centrality between Int farms and N-Int farms is shown.

**Figure 3 pathogens-09-00903-f003:**
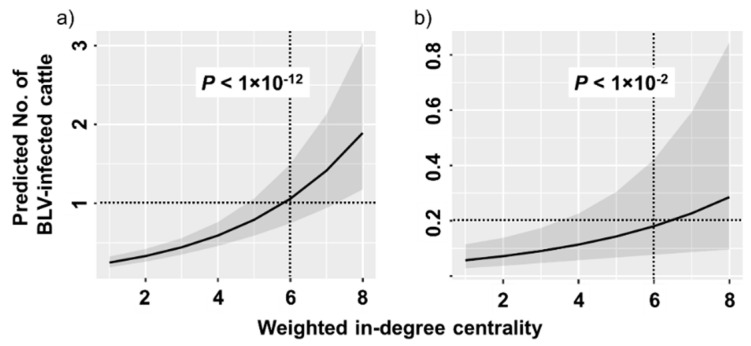
Fit plot of the number of bovine leukemia virus (BLV)-infected cattle among introduced cattle. Subfigure (**a**) depicts the plot based on scenario 1 and subfigure (**b**) depicts the plot based on scenario 2. A fit plot of the generalized linear mixed model describing the predicted number of BLV-infected cattle and the value of weighted in-degree centrality is shown. The line indicates a generalized linear mixed model and the shadow covers a 95% confidence interval.

**Figure 4 pathogens-09-00903-f004:**
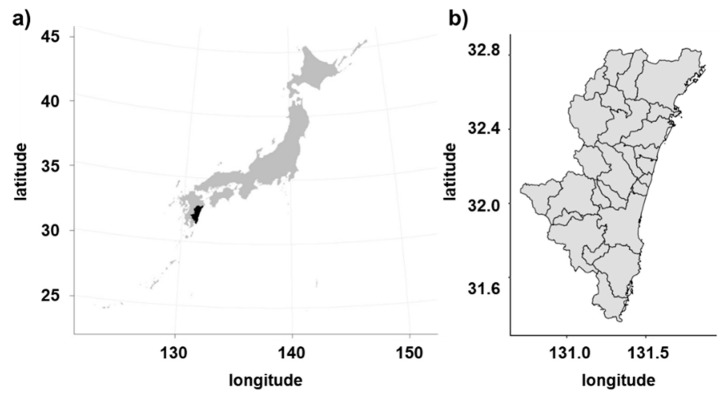
The location of the study area. (**a**) The location of Miyazaki prefecture in Japan. (**b**) Map of Miyazaki prefecture and the border of the cities.

**Figure 5 pathogens-09-00903-f005:**
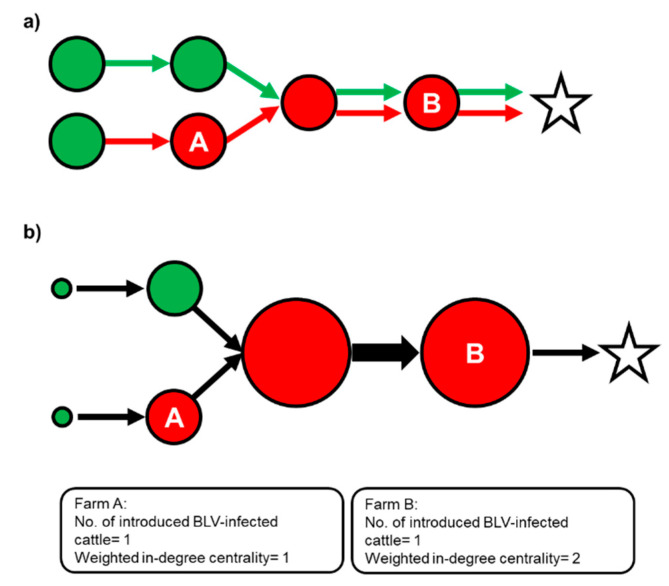
A conceptual diagram of the scenario-based network of cattle movement in scenario 1. Scheme of the construction of a scenario-based network of cattle movement is shown from A to B. A node depicts a premise (a circle is a farm and a star is a slaughterhouse) and a directed tie represents the direction of cattle movement. (**a**) The colors of ties represent bovine leukemia virus (BLV) infectious status of moved cattle based on scenario described in Section 4.2.4. (red tie indicates BLV-infected and green tie indicates BLV-uninfected). The farms that introduced BLV-infected cattle at least once were classified as Int farms (red circle) and the farms without such introduction were classified as N-Int farms (green circle). (**b**) The directed ties from the same sources and directions were merged and weighted by the number of the ties. The size of the nodes and the width of the ties denote weighted in-degree centrality and the weight of ties, respectively. Note that the width of ties toward slaughterhouses was set to one.

**Table 1 pathogens-09-00903-t001:** Proportion of bovine leukemia virus (BLV)- enzyme-linked immunosorbent assay (ELISA)-positive cattle based on sex and breed.

		ELISA-Positive	ELISA-Negative	Total	Proportion of ELISA-Positive (%)
		*n* (heads)	*n* (heads)	*n* (heads)	
Sex	Male	91	262	353	25.8
	Female	107	384	491	21.8
Breed	Japanese black	179	624	803	22.3
	Holstein	10	5	15	66.7
	F1	9	17	26	34.6
Total		198	646	844	23.5

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
