# Peer review of "Quantitative Risk Assessment for the Introduction of Bovine Leukemia Virus-Infected Cattle Using a Cattle Movement Network Analysis"

_pathogens, 2020, doi:10.3390/pathogens9110903_

Round 1

Reviewer 1 Report

This manuscript aims to quantify the farm-level risk of BLV infection based on the translocation of cattle with and without BLV. Using public databases and several assumptions about the rate of BLV positivity in the population of cattle being studied. Following testing of nearly 1,000 animals, ~50% from 2 slaughterhouses and the remainder from farms and markets, it was found that 23.5% of samples were BLV positive. Review of cattle movement revealed that 87% of farms did not receive BLV positive animals. These data were used to create scenarios of cattle movement in order to predict the effect of BLV transmission from the introduction of a BLV-positive animal. The modeling of these scenarios is appreciated and may add to the field’s ability and interest in tracing BLV-positive cattle and the impact this has on farm-level economic considerations. However, the findings from the movement scenario analysis are not surprising, even with the large number of assumptions made and with aspects of in-farm transmission not being considered (line 190). It is suggested that more (anonymized) data that was initially collected from the farms and slaughter houses such as origin of sample, ELISA values, animal attributes such as sex, age, breed, weight be included as a supplementary table to allow readers to recapitulate these methods within different populations and to improve the overall impact of this study.      

Reviewer 2 Report

Please find comments in the attached file.
